# Hederagenin's uric acid-lowering effects in hyperuricemic mice: Mechanistic insights from molecular docking and in vivo analysis

Ping Chen[1]☉, Ya-ni Tian[2]☉, Jing-tao Wang[2], Xiang-lin Yin[3,4], Bao-sheng Guan[3,4], Xue Bai[2,5]*

1 School of Clinical Medicine, Jiamusi University, Jiamusi, China, 2 School of Basic Medicine, Jiamusi University, Jiamusi, China, 3 School of Public Health, Jiamusi University, Jiamusi, China, 4 Heilongjiang Key Laboratory of Gout Research, Jiamusi University, Jiamusi, China, 5 Department of Pharmacy, The Second Affiliated Hospital of Harbin Medical University, Harbin, China

☉ Ping Chen and Ya-ni Tian contributed equally to this study.
* baixue19820526@outlook.com

## Abstract

This study explored the uric acid-lowering effects of hederagenin (HD) through molecular docking analysis and a chronic hyperuricemia (HUA) mouse model. Molecular docking was performed to evaluate HD's interactions key urate-regulating proteins, including xanthine oxidase (XOD), ABCG2, OAT1, URAT1, and GLUT9. To establish a chronic HUA model, mice were fed a yeast-adenine diet supplemented with potassium oxonate. The mice were randomly assigned to six groups: normal control, HUA model control, benzbromarone (BEN) group, and three HD treatment groups at doses of 50, 100, and 200 mg/kg. Serum uric acid (UA) levels, liver and kidney function indicators, XOD activity, and oxidative stress markers were assessed. Histopathological analyses of the liver and kidney were also conducted. In addition, gene and protein expression levels of urate transporters and inflammatory markers were assessed using RT-PCR and Western blotting. The results showed that HD interacts with XOD and urate transporters, significantly reducing serum UA levels and inhibiting XOD activity in HUA model. It also modulated the expression of urate transporter to enhance UA excretion. Moreover, HD protected liver and kidney function by reducing pro-inflammatory cytokine levels and inhibiting the TLR4/Myd88/NF-κB and NLRP3 signaling pathways. These findings suggest HD may serve as a promising therapeutic agent for lowing uric acid and preventing organ damage associated with HUA.

## 1. Introduction

Uric acid (UA) is the final product of purine metabolism in humans. It is mainly produced endogenously through de novo and salvage pathways catalyzed by the enzyme xanthine oxidase (XOD). Approximately 80% of UA originates from these

**Data availability statement:** All relevant data are within the manuscript and its Supporting Information files.

**Funding:** This study was partly supported by the Heilongjiang Province Natural Science Foundation Outstanding Youth Project (YQ2020H001) and the Heilongjiang Province Education Science "13th Five-Year Plan" Project (GBD1317135). The funders had no role in study design, data collection and analysis, decision to publish, or preparation of the manuscript.

**Competing interests:** The authors have declared that no competing interests exist.

internal processes, while the remaining 20% comes from dietary purines [1]. UA is primarily excreted by the kidneys, which account for about two-thirds of the excretion. The intestines and other pathways handle the rest. This excretion process is regulated by several urate transporters, including urate transporter 1 (URAT1), organic anion transporter 1 (OAT1), glucose transporter 9 (GLUT9), and ATP-binding cassette subfamily G member 2 (ABCG2). These transporters control the reabsorption and excretion of UA, ensuring that serum UA levels remain within a normal range under healthy conditions. Disruptions in this balance, whether due to increased UA production or decreased excretion, can lead to hyperuricemia (HUA). Elevated serum UA levels are not only a major risk factor for gout but are also linked to a broad spectrum of metabolic and cardiovascular disorders. These include hypertension, dyslipidemia, insulin resistance, obesity, non-alcoholic fatty liver disease, renal dysfunction, and cardiovascular diseases [2–4]. Given these significant health implications, managing HUA is critical, underscoring the need for effective UA-lowering therapies in both clinical practice and public health.

Current first-line treatments for HUA and gout include XOD inhibitors, such as allopurinol and febuxostat, as well as uricosuric agents like benzbromarone and probenecid. However, these medications often have significant adverse effects, including gastrointestinal irritation, hepatotoxicity, nephrotoxicity, hypersensitivity reactions, and immunogenicity. These side effects can limit their effectiveness and reduce patient compliance. These limitations have driven interest in exploring active components from traditional medicinal plants as potential alternative therapies. Natural compounds derived from these plants often exhibit UA-lowering properties with fewer side effects, making them promising candidates for new HUA treatments. Molecular docking is a computational technique widely used in drug discovery to predict interactions between ligands and receptors based on their geometric, energetic, and chemical complementarity. This method provides valuable insights into the molecular interactions of bioactive compounds with their targets. It aids in identifying potential new therapeutic agents and elucidating their mechanisms of action [5].

Hederagenin (HD) is an oleanane-type pentacyclic triterpenoid found in various traditional medicinal plants. These plants are known for their properties in dispelling wind, removing obstructions, clearing heat, and detoxifying. They belong to families such as Dipsacaceae, Ranunculaceae, Caprifoliaceae, Akebia, Araliaceae, and Paeoniaceae, notably *Dipsacus asperoides, Clematis chinensis, Paeonia lactiflora, and Akebia quinata* [6]. Previous studies have demonstrated that glycoside compounds from these plants, such as the total glycosides of Clematis chinensis and Paeonia lactiflora, can significantly reduce serum UA levels and protect renal function. These effects are attributed to their anti-inflammatory and antioxidant activities [7–9]. These findings suggest that glycoside compounds from traditional Chinese medicinal herbs may hold therapeutic potential for managing HUA.

Beyond its potential UA-lowering effects, HD exhibits a wide range of pharmacological activities. These include antitumor, antidepressant, antibacterial, insecticidal, anti-inflammatory, anti-rheumatic, lipid-lowering, hypoglycemic, and antioxidant

properties [10]. However, the specific relationship between HD and HUA or gout remains underexplored. This study aims to address this gap by confirming the UA-lowering potential of HD and examining its impact on urate transporter regulation. Additionally, it investigates HD's role in modulating inflammatory and oxidative pathways. Unlike previous studies that primarily focused on XOD inhibition, our study provides insights into the broader mechanisms of HD. This include its effects on key signaling pathways such as TLR4/Myd88/NF-κB and NLRP3, offering a more comprehensive understanding of its therapeutic potential in HUA.

The objectives of this study are twofold. First, we identify potential UA-lowering targets of HD through molecular docking. Second, we evaluate its effects on HUA and associated liver and kidney damage in a chronic HUA mouse model. By elucidating the underlying mechanisms of action, this study seeks to provide a foundational theoretical and experimental basis for developing novel therapeutic agents for HUA.

## 2. Materials and methods

### 2.1 Molecular docking

The structure of HD was retrieved from the PubChem database (https://pubchem.ncbi.nlm.nih.gov/). To account for all possible stereoisomers and protonation states, the LigPrep and Epik modules in Schrödinger software (Schrödinger, LLC, USA) were employed. The processed structures were then saved in SDF format for subsequent analysis. The crystal structures of XOD (PDB ID: 3NVY) and ABCG2 (PDB ID: 6FFC) were obtained from the RCSB PDB database (https://www.rcsb.org/). For URAT1, GLUT9, and OAT1, homology models were constructed using the SWISS-MODEL platform (https://swissmodel.expasy.org/) based on sequences retrieved from the UniProt database. The quality of these models was assessed using PROCHECK for Ramachandran plot analysis, Verify3D for model-sequence compatibility, and ProSA-web for overall model quality. Models with over 90% of residues in favored regions were considered suitable for docking studies.

Molecular docking was performed using the Glide module in Schrödinger. The docking grid for XOD was defined around the molybdenum cofactor and flavin adenine dinucleotide, while the ABCG2 grid was centered on the protein-ligand complex. The grid dimensions were set at 20 Å for the outer box and 10 Å for the inner box to fully encompass the entire binding site. Each docking run produced ten conformations per small molecule, followed by energy minimization. The conformation with the highest GlideScore, indicating the best binding affinity, was selected for further analysis. A GlideScore threshold of −5.0 was used to identify strong binding interactions. Both the docking scores and the presence of key interactions, such as hydrogen bonds and hydrophobic contacts, were considered to ensure a thorough evaluation of binding potential. The threshold of −5.0 was chosen based on previous studies that have used similar thresholds to indicate moderate to strong binding affinities in molecular docking studies. For instance, studies by Friesner et al. and others have demonstrated that Glide Scores in the range of −5.0 to −7.0 typically indicate favorable binding interactions, particularly when supported by additional interactions like hydrogen bonds and hydrophobic contacts [11–12].

### 2.2 Drugs and reagents

The catalog numbers and batch details for all drugs, antibodies, and assay kits used in this study are as follows: HD (Nanjing Qiushi Biotechnology Co., Ltd., Catalog No. CAS 465-99-6), potassium oxonate (Shanghai Yuanye Biotechnology Co., Ltd., Catalog No. CAS 2207-75-2), benzbromarone (BEN) (Shanghai Yuanye Biotechnology Co., Ltd., Catalog No. CAS 3562-84-3), sodium carboxymethyl cellulose (Shanghai Yuanye Biotechnology Co., Ltd., Catalog No. CAS 9004-32-4), XOD activity assay kit (Nanjing Jiancheng Bioengineering Institute, Catalog No. A002-1–1), total superoxide dismutase (SOD) activity assay kit (Nanjing Jiancheng Bioengineering Institute, Catalog No. A001-1–2), glutathione peroxidase (GSH-PX) activity assay kit (Nanjing Jiancheng Bioengineering Institute, Catalog No. A005-1–2), and malondialdehyde (MDA) content assay kit (Nanjing Jiancheng Bioengineering Institute, Catalog No. A003-1–2).

## 2.3 Experimental animals

Healthy male Kunming mice, aged 4−6 weeks and weighing 18−22 g, were obtained from the Experimental Animal Department of Harbin Medical University (License No.: SCXK (Hei) 2019−001). The mice were housed in a specific pathogen-free facility under controlled environment conditions, which included a temperature of 22 ± 2°C, relative humidity of 55 ± 5%, and a 12-hour light/dark cycle. They were provided with ad libitum access to standard laboratory chow and sterilized water.

All animal procedures were performed in compliance with the Institutional Animal Care and Use Committee guidelines and approved by the Animal Ethics Committee of Jiamusi University (Approval No.: JMSU-2021101501). To ensure animal welfare, appropriate measures were taken to minimize discomfort and stress during the experiments. Animals were monitored daily for health status, behavior, and any signs of distress.

Prior to interventions, mice were anesthetized via intraperitoneal injection of sodium pentobarbital (3% w/v in sterile saline; 50 mg/kg body weight; Sinopharm Chemical Reagent Co., Ltd.). Anesthesia depth was confirmed by the absence of pedal withdrawal reflexes and stabilized respiratory rates. Supplemental doses (<5 mg/kg) were administered if required to maintain surgical anesthesia.

For blood collection, a standardized retro-orbital puncture technique was executed under deep anesthesia. The periocular area was disinfected with 10% povidone-iodine, and the orbital sinus was accessed using sterile heparinized capillary tubes (1.1 mm diameter). Approximately 0.5 mL of blood was collected per mouse while maintaining a 15° head-down tilt to optimize venous pressure. Hemostasis was achieved by direct compression with sterile gauze for 60 seconds post-procedure.

Euthanasia was performed immediately after blood collection through confirmed secondary methods. Cervical dislocation was conducted by trained personnel using calibrated forceps to ensure rapid atlanto-occipital separation, followed by bilateral thoracotomy to verify cessation of cardiac and respiratory activity. All procedures adhered to the American Veterinary Medical Association Guidelines for the Euthanasia of Animals (2020), with no surviving animals subjected to terminal protocols.

The study adhered to the principles outlined in the Animal Research: Reporting of In Vivo Experiments guidelines and the Institutional Animal Care and Use Committee protocols. Housing conditions, including temperature, humidity, and lighting, were strictly regulated to maintain the animals' physiological and psychological well-being throughout the experimental period.

## 2.4 Preparation and treatment of HUA mouse model

Mice were randomly divided into six groups (n = 6 per group): Normal Control (NC), HUA Model Control (MC), Positive Control (BEN-treated), and three HD treatment groups receiving low (50 mg/kg), medium (100 mg/kg), and high (200 mg/kg) doses of HD. The NC group was fed a standard diet and given saline, while the other groups received a diet containing 10% yeast and 0.1% adenine, along with potassium oxonate (200 mg/kg) to induce HUA. The BEN group was administered 20 mg/kg benzbromarone as a positive control. All treatments were delivered via gavage in a 0.5% sodium carboxymethyl cellulose solution daily at 11:00 AM, with doses adjusted weekly based on body weight measurements.

## 2.5 Sample collection and biochemical analysis

On day 28, one hour after the final gavage, mice were anesthetized, and blood samples were collected from the retro-orbital sinus. Serum was separated by centrifugation at 3500 rpm for 15 minutes at 4°C and stored at −80°C for subsequent biochemical analyses. Mice were euthanized by cervical dislocation, and liver and kidney tissues were collected for histological, transcriptomic, and Western blot analyses. Serum levels of UA, creatinine (Scr), blood urea nitrogen (BUN),

alanine aminotransferase (ALT), and aspartate aminotransferase (AST) were measured using an automated biochemical analyzer (AU5800, Beckman Coulter, USA).

## 2.6 Organ coefficients

The kidneys, liver, thymus, and spleen were carefully isolated, rinsed in ice-cold saline to remove residual blood, and blotted dry with filter paper. Each organ was weighed, and organ coefficients were calculated using the formula:

$$\text{Organ Coefficient (mg/g)} = (\text{Organ Weight (mg)} / \text{Final Body Weight (g)}) * 100$$

## 2.7 XOD activity

Frozen liver tissues were thawed and minced, then homogenized on ice in nine volumes of saline to prepare a 10% liver homogenate. The homogenate was centrifuged, and the supernatant was used to measure XOD activity according to the assay kit's instructions, with results calculated using the provided formula.

## 2.8 Oxidative stress analysis

Liver and kidney tissues were processed to evaluate SOD and GSH-PX activities and MDA concentrations, following the protocols provided with the assay kits. Preliminary experiments were conducted to determine the optimal sample concentrations for these assays.

## 2.9 Histopathological analysis

Liver and kidney tissues were fixed in 10% neutral buffered formalin and processed through dehydration, clearing, paraffin embedding, and sectioning into 4 µm thick slices. Sections were mounted on slides, baked, stained with hematoxylin and eosin (HE), and examined under a microscope. Representative fields were captured using an upright-inverted integrated microscope system (Echo Laboratories, USA).

To objectively assess tissue injury, HE-stained sections were imaged, and the stained area was analyzed using ImageJ software (NIH, USA) to measure optical density for semi-quantitative comparison [13]. Experimental data were analyzed and plotted using GraphPad Prism 8.0.

## 2.10 Transcriptomic analysis

Kidney tissues were homogenized, and RNA was extracted using Trizol reagent (Solarbio Life Sciences, Beijing, China). RNA quality was assessed, and complementary DNA (cDNA) was synthesized by reverse transcription. Quantitative RT-PCR was performed using gene-specific primers on the Thermal Cycler T100 (Bio-Rad, USA) (S1 File). The expression of genes, including OAT1, OAT3, ABCG2, URAT1, and GLUT9, was normalized against GAPDH as an internal control.

## 2.11 Western blotting

Protein samples were extracted from kidney tissues using RIPA lysis buffer (Xi'an Hat Biotechnology Co., Ltd., China). The tissues were homogenized on ice, centrifuged at 12,000 rpm for 10 minutes at 4°C, and the supernatant containing proteins was collected. Protein concentrations were determined using a BCA assay kit (Wuhan Boster Biological Technology Co., Ltd., China). Proteins (20 µg per sample) were separated by 10% SDS-PAGE and transferred to PVDF membranes (Sigma-Aldrich). Membranes were blocked with 5% non-fat milk in TBST (Tris-buffered saline with 0.1% Tween-20) and probed overnight 4°C with primary antibodies. After incubation with secondary antibodies at room temperature, detection was performed using a chemiluminescent detection reagent (Dalian Meilun Biotechnology Co., Ltd., China). Bands

were visualized on a C500 near-infrared imaging system (Azure Biosystems, USA), and band intensities were quantified using ImageJ software (National Institutes of Health, USA), with β-actin serving as a loading control.

## 2.12 Statistical analysis

All experimental data were expressed as mean ± standard error of the mean. Statistical analyses were performed using GraphPad Prism 8.0 software. One-way ANOVA was used to compare quantitative data across multiple groups, followed by Tukey's post-hoc test to determine specific group differences. A $P < 0.05$ was considered statistically significant, and a $P < 0.01$ was considered highly significant. Each experiment was independently replicated three times to ensure the reliability of the results.

## 3. Results

### 3.1 Molecular docking and homology modeling

**Homology modeling results for OAT1, URAT1, and GLUT9.** The reliability of the homology models for OAT1, URAT1, and GLUT9 was validated using Ramachandran plot analysis. For the OAT1 model, 91.91% of the amino acid residues were located within the most favored and allowed regions, meeting the criterion of >90% for a reliable model [14], indicating high accuracy and dependability (Fig 1A). The URAT1 model showed that 90.75% of the residues fell within the favored regions, confirming its reliability and accuracy (Fig 1B). For GLUT9, 98.65% of the residues were within the most favored regions, further validating the high reliability of this model (Fig 1C).

**Molecular docking results of HD with XOD and urate transporters.** Molecular docking studies revealed key interactions between HD and several proteins involved in UA metabolism. For XOD, HD exhibited a docking score of −3.1, forming hydrogen bonds with GLN144 and LYS1228, as well as additional interactions such as hydrogen bonds and salt bridges with ARG426 (Fig 1D). For ABCG2, HD had a docking score of −4.375, primarily interacting through hydrogen bonding with THR435 and additional hydrophobic interactions, contributing to a stable binding within the protein's pocket

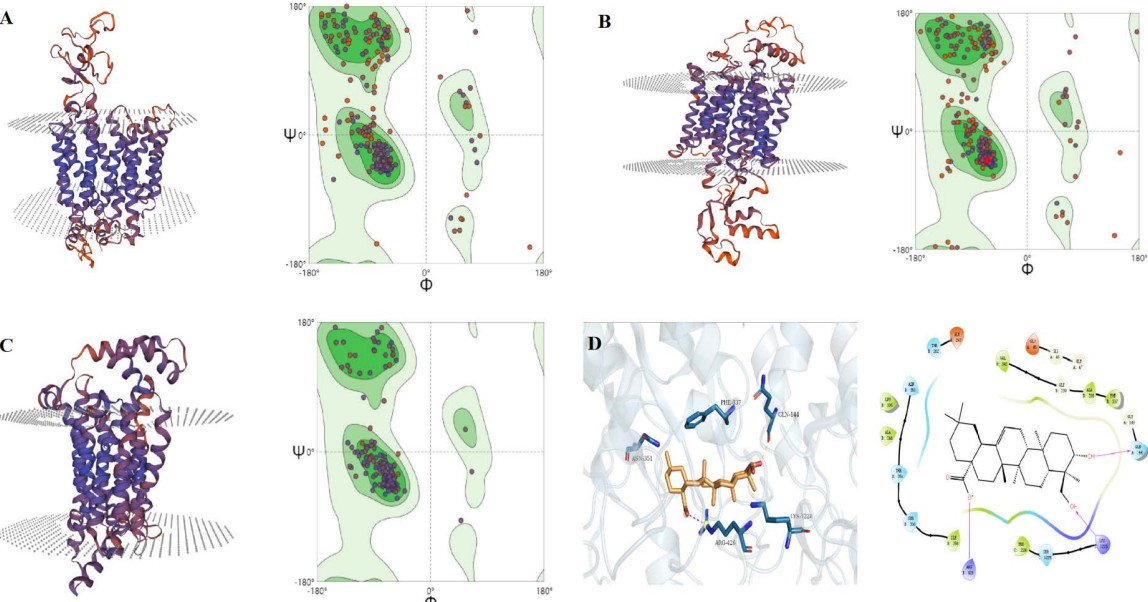

**Fig 1. The modelled structure and Ramachandran plot of OAT1.** (A), URAT1 (B), GLUT9 (C) models, and docked complex of hederagenin (2D and 3D) with XOD (D).

(Fig 2A). With OAT1, HD demonstrated a docking score of −7.2, forming hydrogen bonds with ARG466 and ASN439, along with hydrophobic interactions that enhanced binding within the protein's binding pocket (Fig 2B). For URAT1, HD showed a docking score of −6.558, engaging in hydrogen bonds with THR450 and SER238, as well as a salt bridge with ARG477 (Fig 2C). The docking score for HD with GLUT9 was −6.905, with HD forming hydrogen bonds with TYR71 and TYR327, facilitating its binding within the transporter (Fig 2D).

### 3.2  Effects of HD on serum biochemical parameters in mice

**Serum UA and Liver/Kidney function indicators.**  Chronic HUA induction was confirmed by significantly elevated serum UA levels in the MC group compared to the NC group ($p < 0.01$). Treatment with BEN significantly reduced serum UA levels in the BEN group compared to the MC group ($p < 0.01$). All HD-treated groups also showed a significant reduction in serum UA levels compared to the MC group ($p < 0.01$) (Fig 3). Additionally, the MC group exhibited elevated levels of Scr, BUN, ALT, and AST compared to the NC group ($p < 0.01$). Following BEN treatment, reductions in Scr and BUN ($p < 0.05$) as well as ALT and AST levels ($p < 0.01$) were observed compared to the MC group. Similarly, all HD-treated groups demonstrated significantly lower levels of these biochemical markers compared to the MC group ($p < 0.01$) (Fig 4).

### 3.3  Effects of HD on organ coefficients in HUA mice

**Kidney, Liver, Thymus, and Spleen indices.**  The MC group showed a significant increase in the kidney index compared to the NC group ($p < 0.01$), indicating kidney enlargement. All HD-treated groups exhibited significantly lower kidney indices compared to the MC group ($p < 0.01$) (Fig 5A). No significant differences in liver indices were observed among the experimental groups ($p > 0.05$) (Fig 5B). Additionally, the MC group had significantly higher thymus and spleen indices compared to the NC group ($p < 0.01$). However, these indices were significantly reduced in all HD treatment groups relative to the MC group ($p < 0.01$) (Fig 5C and 5D).

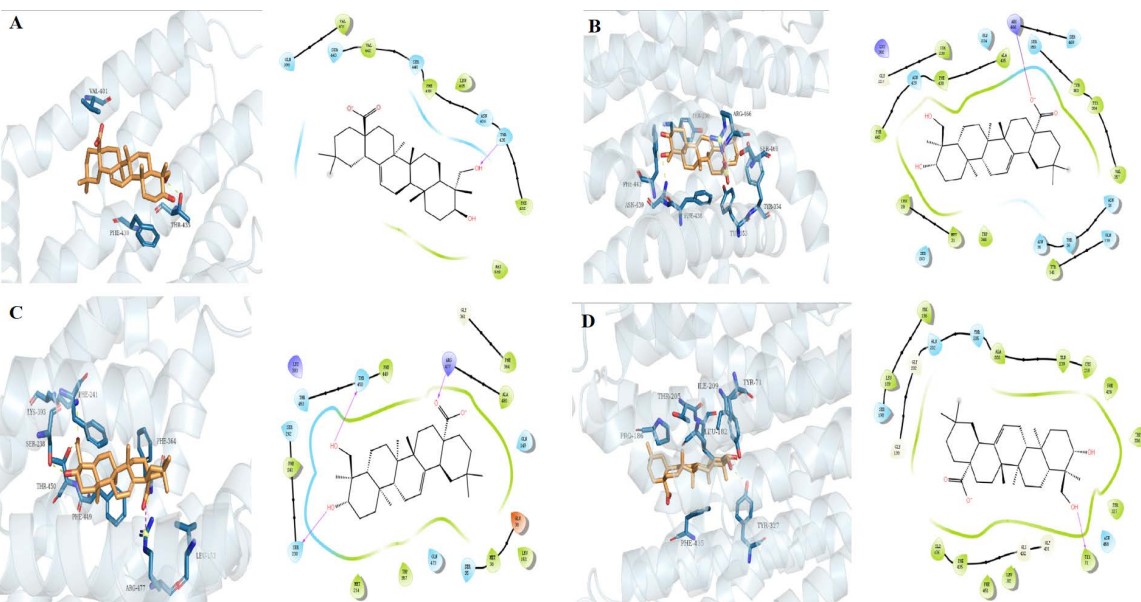

**Fig 2.  Docked complex of hederagenin (2D and 3D) with ABCG2.** (A), OAT1 (B), URAT1 (C), and GLUT9 (D).

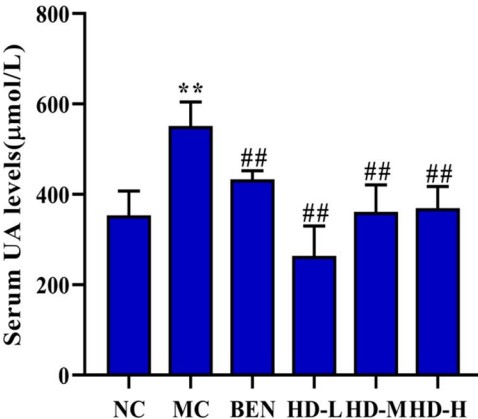

**Fig 3. Effects of HD on serum uric acid levels in mice (\*\*p<0.01 compare with NC group; ##p<0.01 compare with MC group).**

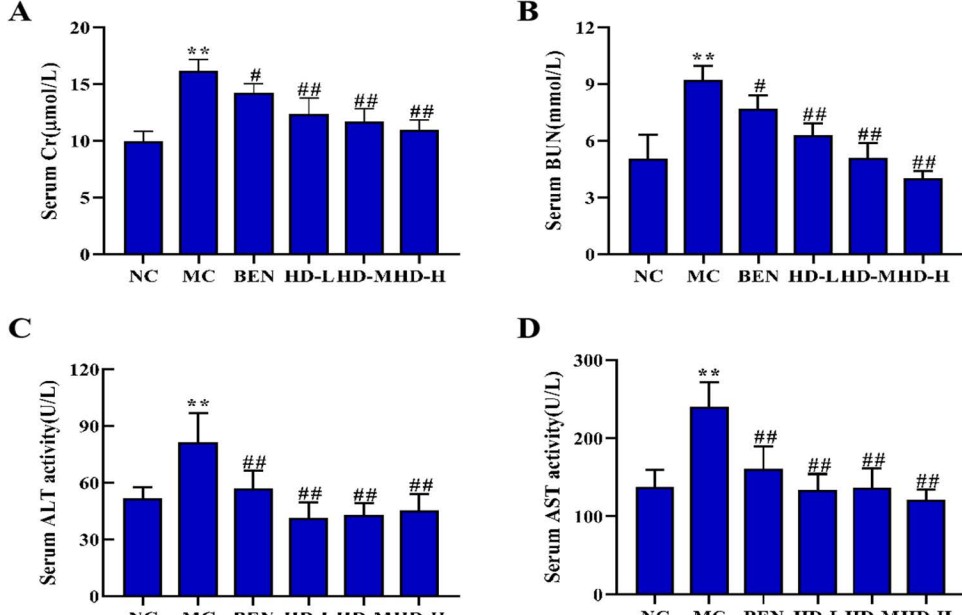

**Fig 4. Effects of HD on serum Cr.** (A), BUN (B), ALT(C), and AST(D) in mice (\*\*p<0.01 compare with NC group; #p<0.05, ##p<0.01 compare with MC group).

### 3.4 Measurement of XOD activity in the Liver of HUA mice

**XOD activity.** Liver XOD activity was significantly higher in the MC group compared to the NC group (p<0.01), confirming increased UA production. Treatment with HD led to a significant, dose-dependent reduction in XOD activity in all HD groups compared to the MC group (p<0.01), with higher doses resulting in greater reductions (Fig 6).

### 3.5 Effects of HD on oxidative stress in the Kidney and Liver

**Oxidative stress markers.** The MC group had significantly elevated MDA levels in both liver and kidney tissues compared to the NC group (p<0.01), indicating increased oxidative stress. Concurrently, the activities of SOD and

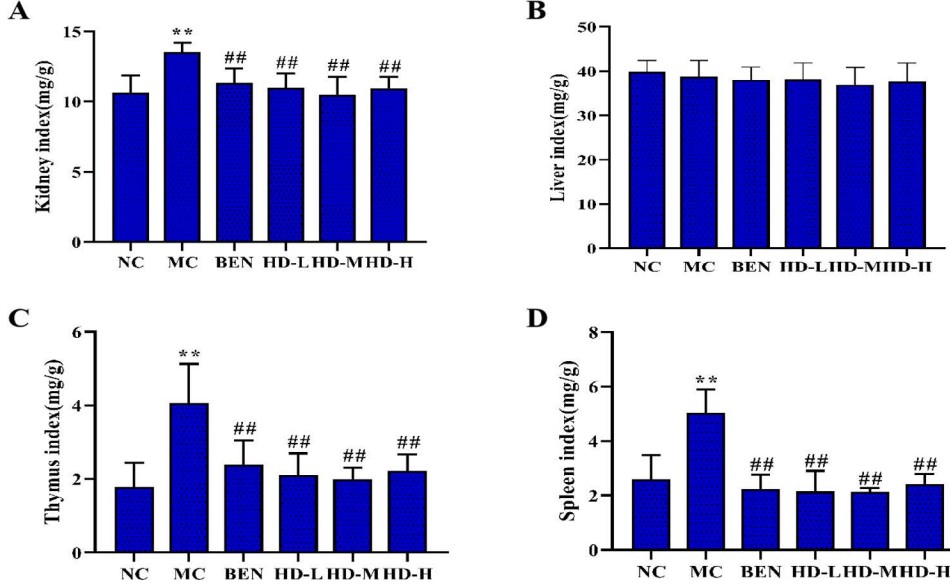

**Fig 5. Effects of HD on organ index in mice.** (A), Kidney index (B), Liver index (C), and Thymus index (D), and Spleen index (**p<0.01 compare with NC group; ##p<0.01 compare with MC group).

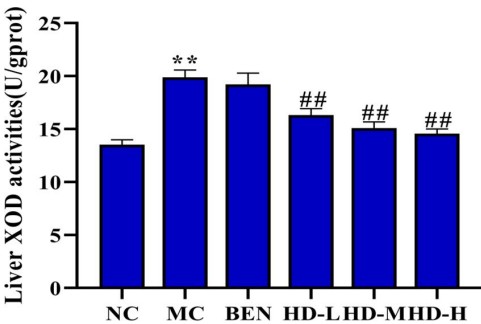

**Fig 6. Effects of HD on liver XOD activities in mice (Note: *p<0.05, **p<0.01 compare with NC group; ##p<0.01 compare with MC group).**

GSH-PX were markedly reduced in the MC group (p<0.01). Treatment with BEN and HD significantly reduced MDA levels and increased SOD and GSH-PX activities in the liver and kidney compared to the MC group (p<0.05) (Fig 7A-7F).

### 3.6 Histopathological examination of Liver and Kidney tissues

**Liver and Kidney tissue analysis.** Histopathological examination of liver tissues revealed that the NC group maintained normal hepatic architecture with intact lobular boundaries (Fig 8). In contrast, the MC group exhibited marked hepatocyte swelling, indistinct blurred lobular boundaries, and abundant inflammatory bodies (*P*<0.01 vs. NC group; Fig 8A and 8B). The BEN group showed partial attenuation of these pathological features, while HD-treated group displayed significant restoration of lobular clarity and inflammatory cells (*P*<0.01 vs.MC group; Fig 8A and 8B), corroborated by semi-quantitative analysis of HE-stained sections (Fig 8B).

Kidney tissues in the NC group maintained regular glomerular morphology and tubular structure (Fig 9). The MC group, however, presented severe glomerular atrophy, tubular dilation, and significant inflammatory cell infiltration (*P*<0.01 vs.

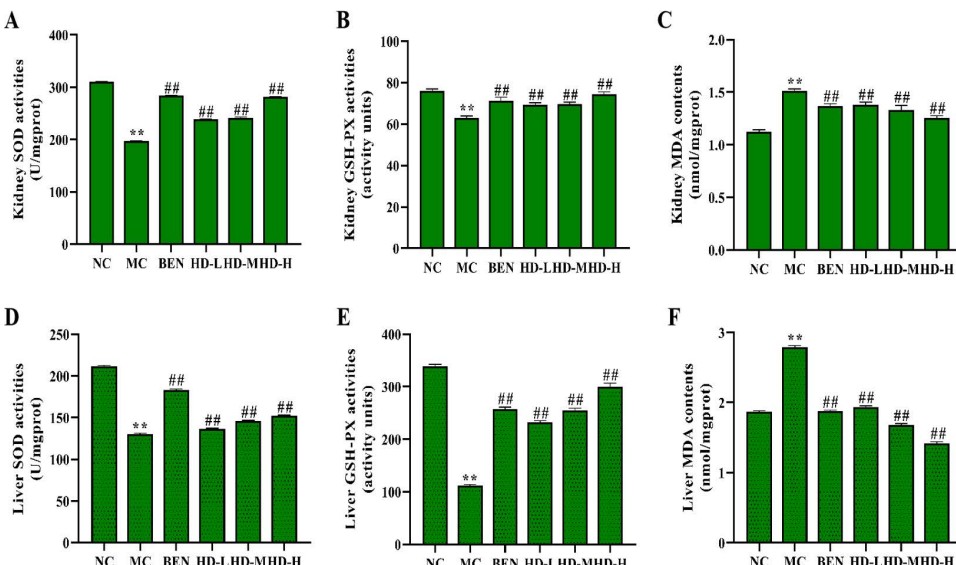

**Fig 7. Effects of HD on oxidative stress indices of kidney and liver in mice on Kidney SOD activities.** (A), Kidney GSH-PX activities (B), Kidney MDA contents (C), Liver SOD activities (D), Liver GSH-PX activities (E), and Liver MDA contents (F).

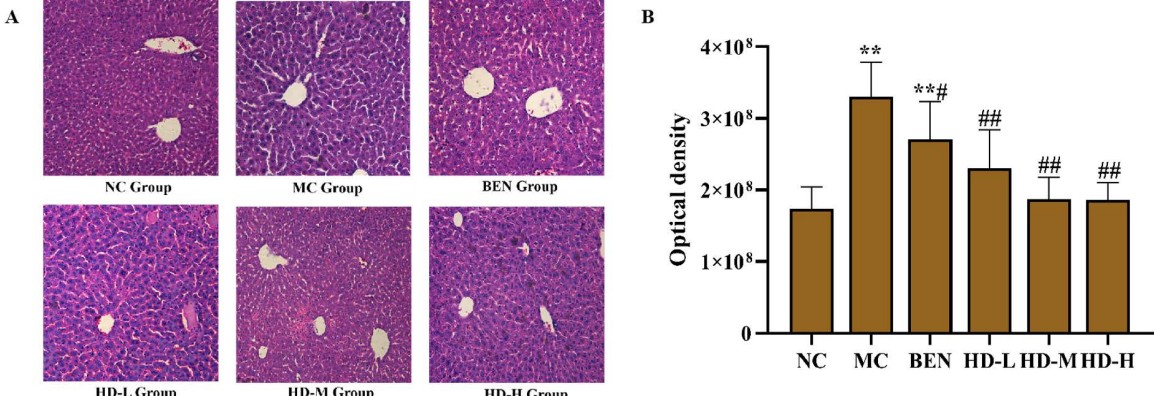

**Fig 8. Histological analyses micrograpahs of liver tissue stained with HE (HE, scale bar 100 μm).** (A), and results of semi-quantitative analysis(**p < 0.01 compare with NC group; ##p < 0.01 compare with MC group) (B).

NC group; Fig 9A and 9B). Although the BEN group exhibited comparable alterations, the HD-treated group showed notable mitigation of these lesions, including diminished tubular injury and inflammatory infiltration (*P* < 0.01 vs.MC group; Fig 9A and 9B), as confirmed by semi-quantitative analysis (Fig 9B).

### 3.7 Effects of HD on mRNA and protein expression levels of Renal UA transporters

**Renal transporter mRNA and protein expression.** In the MC group, the mRNA and protein expression levels of OAT1, OAT3, and ABCG2 were significantly decreased, while URAT1 and GLUT9 were significantly increased compared to the NC group (p < 0.01). Treatment with BEN and HD resulted in significantly increased expression levels of OAT1, OAT3, and ABCG2, and decreased GLUT9 levels (p < 0.01) compared to the MC group. Notably, a significant reduction

in URAT1 expression was observed only in the 50 mg/kg HD group (p < 0.01), consistent with both mRNA and protein findings (Figs 10–12; S2 and S3 Files).

## 3.8 Effects of HD on inflammatory cytokines and signaling pathways in Kidney tissues

**Inflammatory cytokines and TLR4/NLRP3 signaling pathways.** In the MC group, the expression levels of inflammatory cytokines TNF-α, IL-6, and IL-1β were significantly elevated in kidney tissues compared to the NC group (p < 0.01). Treatment with BEN and HD significantly reduced the levels of these cytokines (p < 0.05) (Fig 13; S3 File). Analysis of the TLR4/Myd88/NF-κB pathway showed elevated levels of TLR4, Myd88, and NF-κB p65 proteins in the MC

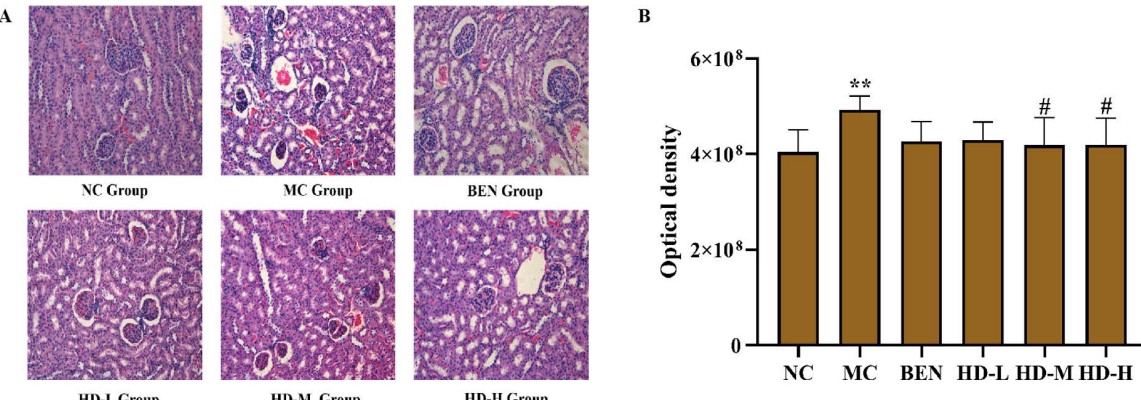

**Fig 9. Histological analyses micrograpahs of renal tissue stained with HE (HE, scale bar 100 μm).** (A), and results of semi-quantitative analysis (**p < 0.01 compare with NC group; ##p < 0.01 compare with MC group) (B).

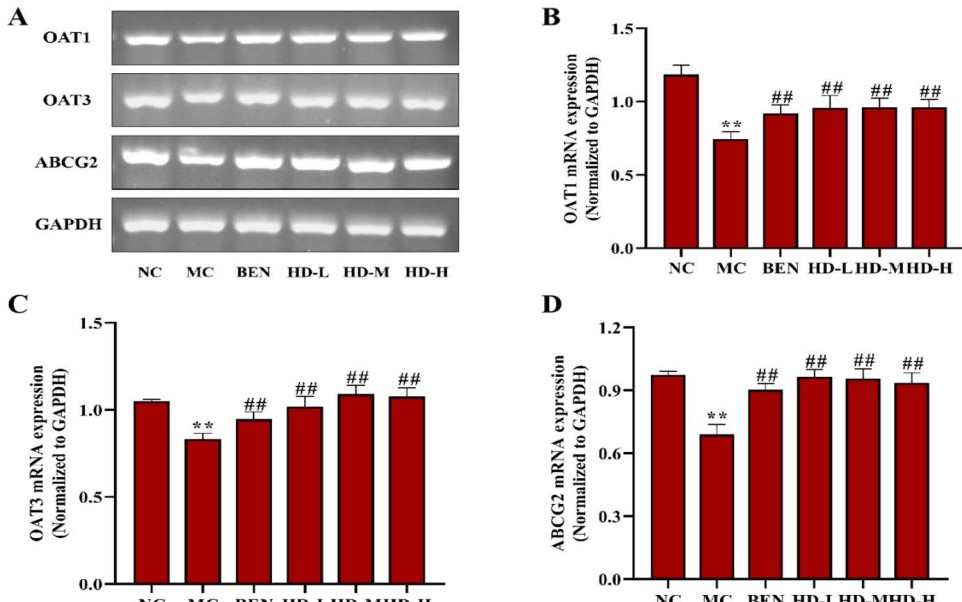

**Fig 10. Effects of HD on mRNA of OAT1, OAT3 and ABCG2 of kidney in mice.** (A), RT-PCR was used to detect the gene expression of OAT1, OAT3 and ABCG2; Results of semi-quantitative analysis of OAT1 (B), OAT3 (C), and ABCG2 (D).

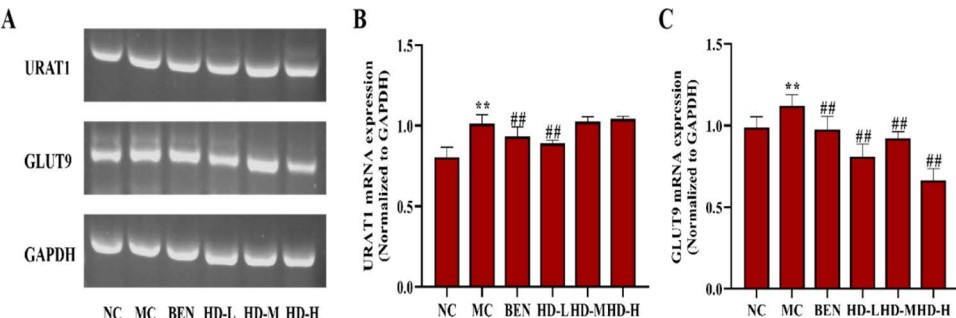

**Fig 11. Effects of HD on mRNA of URAT1 and GLUT9 of kidney in mice as follows: RT-PCR was used to detect the gene expressions of URAT1 and GLUT9.** (A), results of semi-quantitative analysis of URAT1 and GLUT9 (B-C) (**p<0.01 compare with NC group; ##p<0.01 compare with MC group).

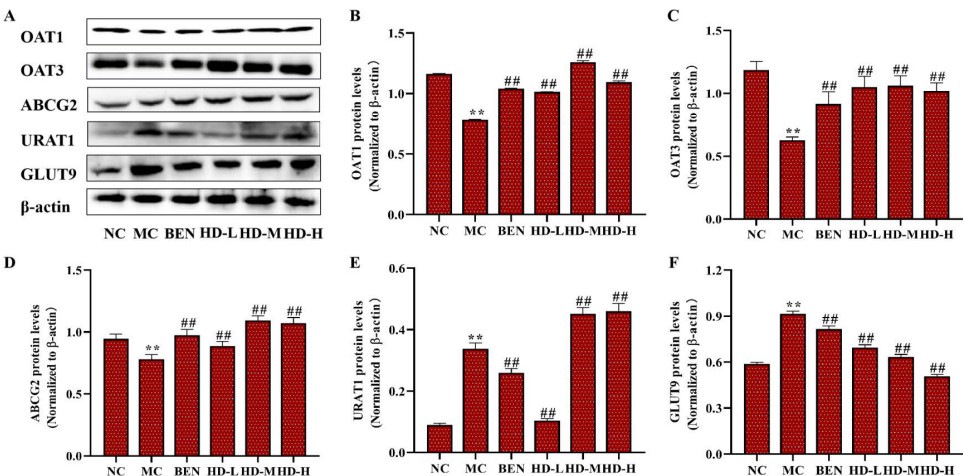

**Fig 12. Effects of HD on the protein expressionsof uric acid transporters of kidney in mice as follows: western blot was used to detect the protein expressions of OAT1, OAT3, ABCG2, URAT1 and GLUT9.** (A), and results of semi-quantitative analysis of OAT1, OAT3, ABCG2, URAT1 and GLUT9 (B-F) (**p<0.01 compare with NC group; ##p<0.01 compare with MC group).

group (p<0.01), which were significantly reduced in the HD-treated groups (p<0.01) (Fig 14; S3 File). Additionally, HD affected the NLRP3 signaling pathway by significantly lowering the expression levels of NLRP3, ASC, and Caspase-1 proteins in the MC group (p<0.01), demonstrating its role in reducing renal inflammation and damage (Fig 15; S3 File).

## 4. Discussion

### 4.1 Role of HUA in disease pathogenesis and the potential of HD

HUA is a major contributor to gout, gouty arthritis, chronic kidney disease (CKD), and cardiovascular diseases (CVD) through multiple molecular pathways. These pathways include the regulation of inflammatory responses, oxidative stress, insulin resistance, endothelial dysfunction, vascular smooth muscle cell proliferation, and activation of the renin-angiotensin system. Elevated UA levels are closely linked to the onset and progression of metabolic syndrome, CVD, and CKD, making UA reduction a critical therapeutic target for these conditions [15]. Our research group has focused on developing UA-lowering therapies and previously identified HD, an oleanane-type pentacyclic triterpenoid,

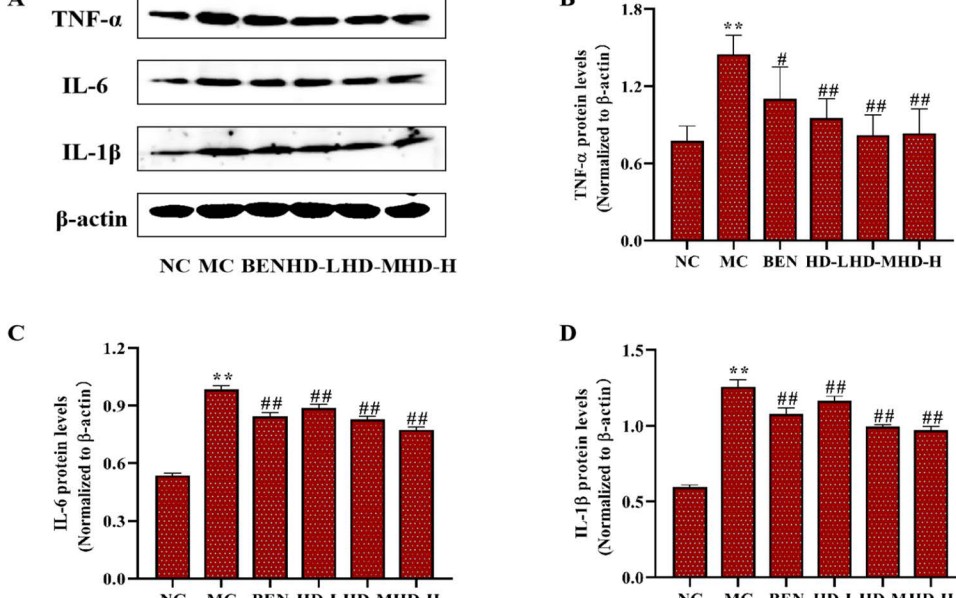

**Fig 13. Effects of HD on inflammatory cytokines in renal tissue homogenatein mice as follows: western blot was used to detect the protein expressions of TNF- α, IL-6 and IL-1β. (A), and results of semi-quantitative analysis of TNF-α, IL-6 and IL-1β (B-D)** ($**p < 0.01$ compare with NC group; $\#p < 0.05$, $\#\#p < 0.01$ compare with MC group).

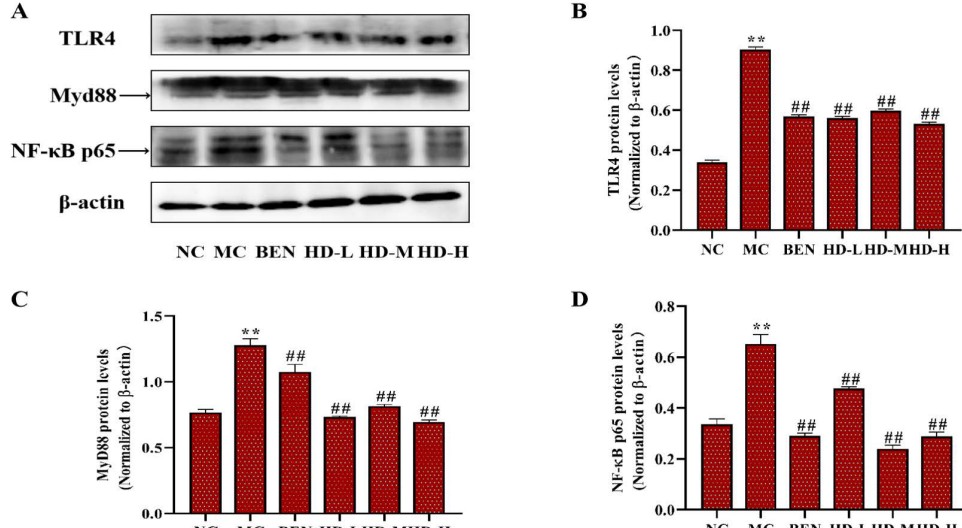

**Fig 14. Expressions of TLR4 pathway related proteins of kidney in mice as follows: western blot was used to detect the protein expressions of TLR4, Myd88 and NF- κB p65. (A), and results of semi-quantitative analysis of TLR4, Myd88 and NF-κB p65 (B-D)** ($**p < 0.01$ compare with NC group; $\#\#p < 0.01$ compare with MC group).

as a compound with anti-hepatocarcinogenic and nephroprotective effects [6]. This study aimed to further explore HD's potential as a UA-lowering agent using molecular docking techniques to investigating its interactions with key targets associated with HUA.

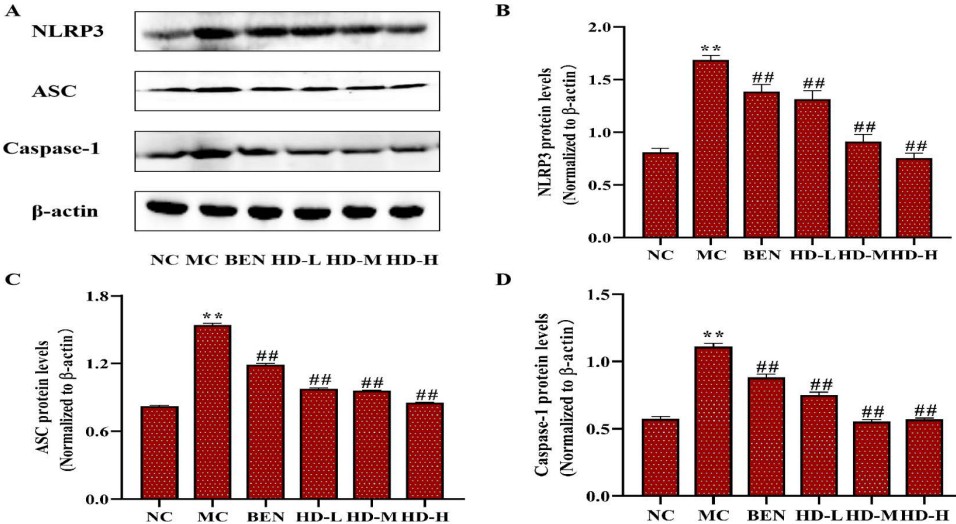

**Fig 15. Expressions of NLRP3 pathway related proteins of kidney in mice as follows: western blot was used to detect the protein expressions of NLRP3, ASC and Casepase-1.** (A), and results of semi-quantitative analysis of NLRP3, ASC and Casepase-1 (B-D) (**p < 0.01 compare with NC group; ##p < 0.01 compare with MC group).

## 4.2 Molecular docking insights into HD's mechanisms of action

Molecular docking studies revealed that HD interacts with several target proteins involved in UA metabolism. Specifically, HD formed hydrogen bonds and salt bridges with XOD at GLN144, LYS1228, and ARG426, suggesting it may inhibit this enzyme. It also binds to ABCG2 via a hydrogen bond with THR435. In addition, HD interacts with OAT1 through hydrogen bonds at ARG466 and ASN439, and with URAT1 through both hydrogen bonds and a salt bridge involving THR450, SER238, and ARG477. For GLUT9, HD forms hydrogen bonds with TYR71 and TYR327. These findings indicate that HD targets key binding sites on these proteins, potentially exerting its UA-lowering effects by modulating both UA synthesis and transport.

## 4.3 Validation of HD's UA-lowering effects in a chronic HUA mouse model

To validate HD's UA-lowering potential and elucidate its underlying mechanisms, we established a chronic HUA mouse model using potassium oxonate, yeast, and adenine. The successful establishment of HUA was confirmed by significantly elevated serum UA levels in the MC group. Treatment with HD significantly reduced serum UA levels, indicating its promising therapeutic effect. Moreover, HD markedly suppressed hepatic XOD activity, suggesting that it lowers serum UA by inhibiting XOD activity. Further analysis revealed that HD increased the mRNA and protein expression of renal UA transporters OAT1, OAT3, and ABCG2, while reducing the expression of GLUT9. Interestingly, only the 50 mg/kg dose of HD effectively inhibited URAT1 expression, whereas higher doses led to its upregulation. This paradoxical effect may result from a compensatory feedback mechanism triggered by sustained serum UA reduction, which in turn stimulates URAT1 expression to maintain UA homeostasis.

The observed URAT1 upregulation at higher HD doses supports the hypothesis of a dose-dependent feedback response to serum UA lowering. This mechanism may counteract the therapeutic effects of HD, emphasizing the importance of dose optimization [16]. Lower doses, such as 50 mg/kg, effectively reduce serum UA without activating compensatory URAT1 expression, while higher doses may diminish efficacy by inducing this response. These findings highlight the need to refine dosing strategies to prevent feedback regulation that could impair treatment outcomes. Future studies

should consider combination therapies aimed at suppressing URAT1 compensation, thereby enhancing the efficacy of HD in HUA management. Based on current results, we propose that HD reduces UA levels by inhibiting hepatic XOD activity and regulating renal UA transporters.

### 4.4 Protective effects of HD on Liver and Kidney function

Elevated serum UA levels are known to impair liver and kidney function [17–20]. Previous studies, including ours, have shown that HD does not exert significant toxic effects on liver and kidney tissues [6]. Moreover, HD has been reported to protect against chemically induced liver injury, reduce renal fibrosis, and alleviate acute kidney damage [21–26]. In this study, we assessed HD's protective effects in chronic HUA mice by measuring serum markers (Scr, BUN, ALT, AST), evaluating organ indices, and performing histopathological analyses. HD treatment significantly improved liver and kidney function and attenuated immune dysfunction caused by elevated serum UA levels. These protective effects are likely attributed to HD's anti-inflammatory and antioxidant properties [27–30]. To explore this hypothesis further, we measured inflammatory cytokines and oxidative stress markers, including SOD, GSH-PX, and MDA, in kidney tissues. HD significantly reduced levels of pro-inflammatory cytokines TNF-α, IL-6, and IL-1β, indicating its potential to alleviate renal inflammation by suppressing these cytokines. In addition, HD enhanced the activities of antioxidant enzymes SOD and GSH-PX while lowering MDA levels in both liver and kidney tissues. These results support the notion that HD protects liver and kidney function by mitigating oxidative stress and inflammation.

### 4.5 Modulation of inflammatory pathways by HD

Further investigation into HD's mechanisms revealed its ability to modulate inflammatory signaling pathways in kidney tissues. HD significantly downregulated key proteins involved in the TLR4/Myd88/NF-κB p65 signaling pathway and the NLRP3 inflammasome. Specifically, HD treatment reduced the expression of TLR4, Myd88, and NF-κB p65, as well as NLRP3, ASC, and Caspase-1. These findings suggest that HD alleviates renal inflammation by inhibiting these pathways, thereby decreasing the release of pro-inflammatory cytokines [27,28,31,32]. This indicates HD's potential to mitigate inflammation-associated renal damage in HUA.

Beyond its anti-inflammatory effects, HD may also influence metabolic signaling pathways such as PI3K/Akt and AMPK, which are central to regulating metabolism and inflammation. The PI3K/Akt pathway is known to control cell growth, survival, and metabolic activity, whereas AMPK functions as a central energy sensor maintaining energy balance. Previous studies have shown that triterpenoids, including HD, can modulate these pathways, producing both anti-inflammatory and metabolic benefits [33–34]. For instance, AMPK activation is associated with reduced inflammation and improved insulin sensitivity, while PI3K/Akt inhibition has been linked to decreased oxidative stress and inflammatory responses [33–34]. Based on this evidence, we hypothesize that HD may exert its uric acid-lowering and organ-protective effects, at least in part, by regulating the PI3K/Akt and AMPK pathways. Future studies should explore the potential interaction between HD and PI3K/Akt/AMPK signaling to further elucidate its therapeutic mechanisms in HUA and related metabolic conditions.

### 4.6 Limitations and future directions

Despite the promising findings, this study has several limitations. The molecular docking results, while supportive, are predictive and require further validation through experimental studies in humans to confirm their clinical relevance. Additionally, the mechanistic pathways explored in this study, including the modulation of urate transporters and inflammatory pathways, represent only part of HD's potential therapeutic actions. Other pathways or compensatory mechanisms might also contribute to HD's effects and were not fully investigated in this study.

Additionally, while our molecular docking results provide valuable insights into HD's interactions with key proteins involved in UA metabolism, further experimental validation through in vitro enzymatic assays and site-directed mutagenesis would strengthen these findings. Specifically, future studies could investigate the direct inhibition of XOD activity by HD

and explore the effects of specific mutations in the binding sites of URAT1, GLUT9, and ABCG2 on HD's binding affinity. These approaches would offer more robust mechanistic insights into HD's UA-lowering effects and its potential as a therapeutic agent for HUA. Future research should focus on clinical trials and broader mechanistic studies to comprehensively elucidate HD's therapeutic potential and safety profile in managing HUA.

To address current limitations and better elucidate the therapeutic potential of HD, future research should prioritize several key directions. First, well-designed clinical trials are essential. A Phase I clinical trial involving healthy volunteers and patients with HUA should be conducted to evaluate the safety, tolerability, pharmacokinetics, and optimal dosing of HD. Such a study would provide foundational data on HD's bioavailability and potential adverse effects. Subsequently, Phase II trials should assess HD's efficacy in lowering serum uric acid levels and improving clinical symptoms in patients with HUA or gout. Second, the inclusion of comprehensive biomarker assessments will be critical to understanding HD's mechanisms of action. Specifically, monitoring changes in pro-inflammatory cytokines (e.g., TNF-α, IL-6, IL-1β) and oxidative stress markers (e.g., SOD, GSH-PX, MDA) could offer valuable insights into HD's anti-inflammatory and antioxidant properties. These biomarkers may also help identify patient subgroups that are more likely to benefit from HD therapy. Third, considering the potential for compensatory responses—such as URAT1 upregulation at higher HD doses—future studies should investigate combination therapy strategies. Co-administration with established urate-lowering agents like allopurinol or febuxostat may produce synergistic effects by modulating multiple targets within uric acid metabolism and excretion pathways, while potentially reducing adverse compensatory feedback. Fourth, to address pharmacokinetic limitations such as poor solubility and low bioavailability, research should explore advanced drug delivery technologies. Nanoparticle-based systems and liposomal formulations have demonstrated potential in enhancing the stability and bioefficacy of triterpenoids and may significantly improve the therapeutic profile of HD. Collectively, these proposed directions aim to advance the clinical translation of HD, optimize its application in HUA management, and provide a more comprehensive understanding of its efficacy and safety profile.

### 4.7  Clinical relevance and translational potential

While the current study highlights the efficacy of HD in a hyperuricemic mouse model, its clinical relevance and translational potential warrant further investigation. Previous research on triterpenoids, including HD, has indicated promising bioavailability and pharmacokinetic properties in animal models; however, human data remain scarce. Studies in rats have reported moderate oral bioavailability of HD, with peak plasma concentrations occurring within 30 minutes post-administration [35–36]. Nevertheless, its limited solubility and rapid metabolic clearance may present challenges for clinical application. To address these limitations, future studies could investigate advanced drug delivery systems, such as nanoparticle-based carriers or liposomal formulations, which have demonstrated potential in enhancing the solubility, stability, and bioavailability of triterpenoids [36].

Regarding safety, preclinical studies have shown that HD exhibits a favorable toxicity profile, with no significant adverse effects observed at therapeutic doses [37–38]. However, potential side effects, including gastrointestinal disturbances or hepatotoxicity, should be rigorously evaluated in human trials. Related studies on structurally similar triterpenoids, such as oleanolic acid and ursolic acid, have reported anti-inflammatory and antioxidant benefits with minimal adverse effects, suggesting that HD may possess a comparable safety profile [36].

Translational challenges include the necessity for well-designed clinical trials to evaluate HD's efficacy and safety in human populations, particularly among patients with HUA and gout. Furthermore, exploring combination therapies with established urate-lowering agents, such as allopurinol or febuxostat, could enhance therapeutic outcomes and mitigate compensatory mechanisms, such as URAT1 upregulation. Future research should also examine the potential integration of HD into traditional Chinese medicine formulations, which may offer synergistic effects and improve patient adherence.

## 5. Conclusion

This study demonstrates that HD effectively reduces serum UA by dual mechanisms: inhibiting hepatic XOD activity to decrease UA synthesis, and modulating urate transporters to enhance UA excretion. HD achieves this by downregulating reabsorption transporters URAT1 and GLUT9 and upregulating secretion transporters OAT1, OAT3, and ABCG2, which collectively promote UA elimination. Additionally, HD protects against liver and kidney damage in HUA, largely through its anti-inflammatory and antioxidant effects. It reduces inflammation by inhibiting TLR4/Myd88/NF-κB and NLRP3 pathways, leading to lower pro-inflammatory cytokine levels, while strengthening antioxidant defenses to minimize oxidative stress. Thus, HD emerges as a promising agent for managing HUA, offering UA-lowering and organ-protective benefits. Further research is encouraged to explore HD's clinical potential for HUA and related complications.

## Supporting information

**S1 File. PCR primer sequences and protocols.**
(DOCX)

**S2 File. PCR image data.**
(ZIP)

**S3 File. Western blot image data.**
(7Z)

## Author contributions

**Conceptualization:** Ping Chen, Ya-ni Tian, Jing-tao Wang, Bao-sheng Guan, Xue Bai.

**Data curation:** Ping Chen, Ya-ni Tian, Xiang-lin Yin, Xue Bai.

**Formal analysis:** Jing-tao Wang.

**Funding acquisition:** Xue Bai.

**Investigation:** Xue Bai.

**Methodology:** Ping Chen.

**Project administration:** Xue Bai.

**Resources:** Ping Chen, Ya-ni Tian, Xiang-lin Yin, Bao-sheng Guan.

**Software:** Jing-tao Wang.

**Supervision:** Xue Bai.

**Validation:** Ping Chen, Ya-ni Tian, Jing-tao Wang, Xiang-lin Yin, Bao-sheng Guan, Xue Bai.

**Visualization:** Ping Chen, Ya-ni Tian, Jing-tao Wang, Xiang-lin Yin, Bao-sheng Guan, Xue Bai.

**Writing – original draft:** Ping Chen, Ya-ni Tian, Jing-tao Wang, Xiang-lin Yin, Bao-sheng Guan, Xue Bai.

**Writing – review & editing:** Ping Chen, Ya-ni Tian, Jing-tao Wang, Xiang-lin Yin, Bao-sheng Guan, Xue Bai.

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
