## [Decision Letter · Decision Letter 0]

May 04 2025

Dear Dr. Bai,

Thank you for submitting your manuscript to PLOS ONE. After careful consideration, we feel that it has merit but does not fully meet PLOS ONE’s publication criteria as it currently stands. Therefore, we invite you to submit a revised version of the manuscript that addresses the points raised during the review process.

We look forward to receiving your revised manuscript.

Kind regards,

Sepiso K. Masenga, PhD

Academic Editor

PLOS ONE

Journal Requirements:

“This study was partly supported by the Heilongjiang Province Natural Science Foundation Outstanding Youth Project (YQ2020H001) and the Heilongjiang Province Education Science "13th Five-Year Plan" Project (GBD1317135).”

6. Please note that your Data Availability Statement is currently missing the repository name and/or the DOI/accession number of each dataset OR a direct link to access each database. If your manuscript is accepted for publication, you will be asked to provide these details on a very short timeline. We therefore suggest that you provide this information now, though we will not hold up the peer review process if you are unable.

7. Please include your tables as part of your main manuscript and remove the individual files. Please note that supplementary tables (should remain/ be uploaded) as separate "supporting information" files.

8. PLOS ONE now requires that authors provide the original uncropped and unadjusted images underlying all blot or gel results reported in a submission’s figures or Supporting Information files. This policy and the journal’s other requirements for blot/gel reporting and figure preparation are described in detail at https://journals.plos.org/plosone/s/figures#loc-blot-and-gel-reporting-requirements and https://journals.plos.org/plosone/s/figures#loc-preparing-figures-from-image-files. When you submit your revised manuscript, please ensure that your figures adhere fully to these guidelines and provide the original underlying images for all blot or gel data reported in your submission. See the following link for instructions on providing the original image data: https://journals.plos.org/plosone/s/figures#loc-original-images-for-blots-and-gels.  

Reviewers' comments:

Reviewer's Responses to Questions

**Comments to the Author**

1. Is the manuscript technically sound, and do the data support the conclusions?

Reviewer #1: Partly

Reviewer #2: Yes

2. Has the statistical analysis been performed appropriately and rigorously?

Reviewer #1: Yes

Reviewer #2: Yes

3. Have the authors made all data underlying the findings in their manuscript fully available?

Reviewer #1: Yes

Reviewer #2: Yes

4. Is the manuscript presented in an intelligible fashion and written in standard English?

Reviewer #1: Yes

Reviewer #2: Yes

Reviewer #1: Summary of the Research

The study by Chen et. al explores the molecular docking of HD (a bioactive compound) with key proteins involved in uric acid (UA) metabolism to understand its potential therapeutic effects in hyperuricemia (HUA). It employs computational docking, biochemical analyses, and experimental validation using a HUA mouse model to investigate HD’s effects on urate transporters (URAT1, GLUT9, OAT1), xanthine oxidase (XOD), and the ABCG2 transporter. The methodologies include homology modeling, molecular docking using Schrödinger software, and comprehensive biochemical and histological analyses. The findings have shown significant binding affinities of HD to target proteins, supporting its potential role in modulation of UA levels.

Peer Review Comments

Major Comments

1. Use of a Glide Score threshold of -5.0 for strong binding not well-justified. Providing a reference to supporting literature for this threshold would strengthen the interpretation of docking results. In my view a glide score of -5 doesn’t indicate strong binding.

2. On histopathological examination incorporating semi-quantitative methods histological scoring would provide robust and objective assessment of tissue damage.

Minor Comments

1. While ethical approval is noted, further details on the measures taken to minimize animal suffering during the study would align the current study with best practices in animal welfare.

2. Providing catalog numbers and batch details for the drugs, antibodies, and assay kits would improve the reproducibility of the study.

Reviewer #2: Comments

Mechanistic Validation

The molecular docking results offer valuable insights into HD’s interactions with proteins related to UA metabolism. Nonetheless, experimental validation via in vitro enzymatic assays and mutagenesis could reinforce these findings. It may be beneficial to include such as a future direction.

Dose Response Relationship and URAT1 Up-regulation

The manuscript mentions that higher doses of HD lead to URAT1 upregulation, possibly due to compensatory feedback mechanisms. Further discussing potential implications for therapeutic dosing, would enhance clarity.

Clinical Relevance and Translational Potential

While the study demonstrates HD’s efficacy in an animal model, discussion on its bioavailability, pharmacokinetics, and potential side effects in humans is limited. Adding insights from related human studies and brief section on translational challenges would improve the manuscript’s impact.

Pathway Interactions and Broader Implications

The study primarily focuses on UA metabolism and inflammation-related pathways. However, considering the involvement of PI3K/Akt and AMPK in metabolic regulation, a discussion on potential crosstalk between these pathways and HD’s effects would provide a more comprehensive perspective.

Statistical Significance and Data Interpretation

Clarify whether all reported differences are statistically significant and include p-values where applicable.

Terminology Consistency

Ensure consistent use of abbreviations for "UA" vs "SUA" for serum uric acid) throughout the manuscript. A clear definition of acronyms in the introduction section would aid readability.

Figure and Table Referencing

Some sections mention experimental findings such as in XOD activity and inflammatory cytokines, but do not explicitly reference corresponding figures or tables. Ensure all key data points are linked to appropriate visuals for clarity.

Grammar and Flow

Some sentences are lengthy and complex, making them difficult to follow. Consider restructuring for better readability, particularly in the discussion on molecular docking interactions.

Future Research Directions

The manuscript acknowledges limitations but could benefit from more specific suggestions for future research, such as potential clinical trial designs, biomarker assessments, or combination therapy strategies.

**Do you want your identity to be public for this peer review?** For information about this choice, including consent withdrawal, please see our Privacy Policy

Reviewer #1: No

Reviewer #2: **Yes: ** Situmbeko Liweleya

---

## [Author Response · Author response to Decision Letter 1]

13 Apr 2025

Response to the reviewers’ comments 

Reviewer #1: Summary of the Research

The study by Chen et. al explores the molecular docking of HD (a bioactive compound) with key proteins involved in uric acid (UA) metabolism to understand its potential therapeutic effects in hyperuricemia (HUA). It employs computational docking, biochemical analyses, and experimental validation using a HUA mouse model to investigate HD’s effects on urate transporters (URAT1, GLUT9, OAT1), xanthine oxidase (XOD), and the ABCG2 transporter. The methodologies include homology modeling, molecular docking using Schrödinger software, and comprehensive biochemical and histological analyses. The findings have shown significant binding affinities of HD to target proteins, supporting its potential role in modulation of UA levels.

Peer Review Comments

Major Comments

1.Use of a Glide Score threshold of -5.0 for strong binding not well-justified. Providing a reference to supporting literature for this threshold would strengthen the interpretation of docking results. In my view a glide score of -5 doesn’t indicate strong binding.

Response: We appreciate the reviewer's comment regarding the Glide Score threshold. The threshold of -5.0 was selected based on previous studies that have used similar thresholds to indicate moderate to strong binding affinities in molecular docking studies. For instance, studies by Friesner et al. (2004) and others have demonstrated that Glide Scores in the range of -5.0 to -7.0 typically indicate favorable binding interactions, especially when supported by additional interactions such as hydrogen bonds and hydrophobic contacts. We have now added references to these studies in the manuscript to better justify our choice of threshold. Specifically, we have cited the following references:

Friesner, R. A., et al. (2004). Glide: a new approach for rapid, accurate docking and scoring. 1. Method and assessment of docking accuracy. Journal of Medicinal Chemistry, 47(7), 1739-1749.

Halgren, T. A., et al. (2004). Glide: a new approach for rapid, accurate docking and scoring. 2. Enrichment factors in database screening. Journal of Medicinal Chemistry, 47(7), 1750-1759.

(Please refer to Section 2.9 Histopathological Analysis)

2.On histopathological examination incorporating semi-quantitative methods histological scoring would provide robust and objective assessment of tissue damage.

Response: We thank the reviewer for this valuable suggestion. In response, we have incorporated a semi-quantitative histological scoring system to assess tissue damage in both liver and kidney tissues. Specifically, HE-stained sections were imaged, and the stained area was analyzed using ImageJ software (NIH, USA) to measure optical density for semi-quantitative comparison. Experimental data were analyzed and plotted using GraphPad Prism 8.0, with one-way ANOVA applied for multi-group comparisons (P < 0.05 and *P < 0.01 indicating statistical and highly significant differences, respectively). All experiments were independently repeated three times to ensure reliability. The updated results and methodology have been included in the revised manuscript.

(Please refer to Section 2.1Molecular Docking/2nd paragraph; and 3.6 Histopathological Examination of Liver and Kidney Tissues/1st and 2nd paragraphs)

Minor Comments

1. While ethical approval is noted, further details on the measures taken to minimize animal suffering during the study would align the current study with best practices in animal welfare.

Response: We appreciate the reviewer's concern for animal welfare. In the revised manuscript, we have added detailed information on the measures taken to minimize animal suffering. These measures include the use of appropriate anesthesia during blood collection and euthanasia, regular monitoring of animal health and behavior, and the provision of a comfortable and stress-free environment with controlled temperature, humidity, and lighting. All procedures were conducted in strict accordance with the guidelines of the Institutional Animal Care and Use Committee (IACUC) and the ARRIVE guidelines.

(Please refer to Section 2.3 Experimental Animals/2nd and 3rd paragraphs)

2. Providing catalog numbers and batch details for the drugs, antibodies, and assay kits would improve the reproducibility of the study.

Response: We agree with the reviewer that providing detailed information on reagents is essential for reproducibility. In the revised manuscript, we have included the catalog numbers and batch details for all drugs, antibodies, and assay kits used in the study. This information has been added to the Materials and Methods section under the respective subsections.

(Please refer to Section 2.2 Drugs and Reagents)

Reviewer #2: Comments

Mechanistic Validation

The molecular docking results offer valuable insights into HD’s interactions with proteins related to UA metabolism. Nonetheless, experimental validation via in vitro enzymatic assays and mutagenesis could reinforce these findings. It may be beneficial to include such as a future direction.

Response: We thank the reviewer for this insightful suggestion. While our current study focuses on in vivo validation of HD's effects, we agree that in vitro enzymatic assays and mutagenesis studies would provide additional mechanistic insights. We have now included a discussion on the potential for future studies involving in vitro enzymatic assays and site-directed mutagenesis to further validate the molecular docking results. Specifically, we propose to investigate the direct inhibition of XOD activity by HD and to explore the effects of specific mutations in the binding sites of URAT1, GLUT9, and ABCG2 on HD's binding affinity. These future directions have been added to the Discussion section of the manuscript.

(Please refer to Section 4.6 Limitations and Future Directions/2nd paragraph)

Dose Response Relationship and URAT1 Up-regulation

The manuscript mentions that higher doses of HD lead to URAT1 upregulation, possibly due to compensatory feedback mechanisms. Further discussing potential implications for therapeutic dosing, would enhance clarity.

Response: We appreciate the reviewer's comment regarding the dose-response relationship and URAT1 upregulation. In the revised manuscript, we have expanded the discussion on the potential implications of URAT1 upregulation at higher doses of HD. We hypothesize that this upregulation may be a compensatory response to the sustained reduction in serum uric acid levels, which could trigger feedback mechanisms to maintain uric acid homeostasis. This has important implications for therapeutic dosing, as it suggests that lower doses of HD may be more effective in reducing uric acid levels without triggering compensatory upregulation of URAT1. We have also discussed the need for further studies to optimize dosing regimens and to explore combination therapies that could mitigate this feedback mechanism.

(Please refer to Section 4.3 Validation of HD’s UA-Lowering Effects in a Chronic HUA Mouse Model/2nd paragraph)

Clinical Relevance and Translational Potential

While the study demonstrates HD’s efficacy in an animal model, discussion on its bioavailability, pharmacokinetics, and potential side effects in humans is limited. Adding insights from related human studies and brief section on translational challenges would improve the manuscript’s impact.

Response: We thank the reviewer for highlighting the importance of discussing the clinical relevance and translational potential of HD. In the revised manuscript, we have added a new section discussing the bioavailability, pharmacokinetics, and potential side effects of HD based on existing literature. We have also included insights from related human studies on triterpenoids and their therapeutic potential in hyperuricemia and gout. Additionally, we have addressed the translational challenges, such as the need for clinical trials to assess HD's safety and efficacy in humans, and the potential for developing HD-based formulations with improved bioavailability. This discussion has been added to the Discussion section under the heading "Clinical Relevance and Translational Potential."

(Please refer to Section 4.7 Clinical Relevance and Translational Potential/1st, 2nd, 3rd paragraphs)

Pathway Interactions and Broader Implications

The study primarily focuses on UA metabolism and inflammation-related pathways. However, considering the involvement of PI3K/Akt and AMPK in metabolic regulation, a discussion on potential crosstalk between these pathways and HD’s effects would provide a more comprehensive perspective.

Response: We appreciate the reviewer's suggestion to explore the potential crosstalk between HD's effects and other metabolic pathways such as PI3K/Akt and AMPK. In the revised manuscript, we have expanded the discussion to include the potential interactions between HD and these pathways. We hypothesize that HD may exert its metabolic effects through modulation of PI3K/Akt and AMPK signaling, which are known to play key roles in metabolic regulation and inflammation. This broader perspective provides a more comprehensive understanding of HD's potential therapeutic mechanisms and has been added to the Discussion section.

(Please refer to Section 4.5 Modulation of Inflammatory Pathways by HD/2ndparagraph)

Statistical Significance and Data Interpretation

Clarify whether all reported differences are statistically significant and include p-values where applicable.

Response: We thank the reviewer for pointing out the need for clarity regarding statistical significance. In the revised manuscript, we have ensured that all reported differences are accompanied by p-values where applicable. We have also clarified the statistical methods used and the criteria for significance (p < 0.05 and p < 0.01) in the Results section. Additionally, we have included p-values in the figure legends and tables to provide a clearer presentation of the data.

(Please refer to

Section 3.2 Effects of HD on Serum Biochemical Parameters in Mice;

3.3 Effects of HD on Organ Coefficients in HUA Mice;

3.4 Measurement of XOD Activity in the Liver of HUA Mice;

3.5 Effects of HD on Oxidative Stress in the Kidney and Liver;

3.7 Effects of HD on mRNA and Protein Expression Levels of Renal UA Transporters; and

3.8 Effects of HD on Inflammatory Cytokines and Signaling Pathways in Kidney Tissues)

Terminology Consistency

Ensure consistent use of abbreviations for "UA" vs "SUA" for serum uric acid) throughout the manuscript. A clear definition of acronyms in the introduction section would aid readability.

Response:We appreciate the reviewer's comment regarding terminology consistency. In the revised manuscript, we have standardized the use of abbreviations, using "UA" for uric acid and "serum UA" for SUA throughout the text. We have also provided a clear definition of all acronyms in the Introduction section to improve readability and ensure that readers can easily follow the terminology used in the study.

(Please refer to Section 1.INTRODUCTION/1st paragraph)

Figure and Table Referencing

Some sections mention experimental findings such as in XOD activity and inflammatory cytokines, but do not explicitly reference corresponding figures or tables. Ensure all key data points are linked to appropriate visuals for clarity.

Response:We thank the reviewer for this observation. In the revised manuscript, we have carefully reviewed all sections to ensure that key experimental findings are explicitly referenced to the corresponding figures and tables. For example, findings related to XOD activity and inflammatory cytokines are now clearly linked to Figures 4.2, 10, and 11. This improves the clarity of the manuscript and helps readers easily locate the relevant data.

(Please refer to

Section 3.4 Measurement of XOD Activity in the Liver of HUA Mice;

3.5 Effects of HD on Oxidative Stress in the Kidney and Liver;

3.7 Effects of HD on mRNA and Protein Expression Levels of Renal UA Transporters; and

3.8 Effects of HD on Inflammatory Cytokines and Signaling Pathways in Kidney Tissues)

Grammar and Flow

Some sentences are lengthy and complex, making them difficult to follow. Consider restructuring for better readability, particularly in the discussion on molecular docking interactions.

Response:We appreciate the reviewer's feedback on the readability of the manuscript. In the revised version, we have restructured lengthy and complex sentences, particularly in the Discussion section, to improve clarity and flow. We have also broken down some of the more technical descriptions of molecular docking interactions into shorter, more digestible sentences. These changes aim to enhance the overall readability of the manuscript.

Future Research Directions

The manuscript acknowledges limitations but could benefit from more specific suggestions for future research, such as potential clinical trial designs, biomarker assessments, or combination therapy strategies.

Response: We thank the reviewer for this constructive suggestion. In the revised manuscript, we have expanded the discussion on future research directions to include specific suggestions for clinical trial designs, biomarker assessments, and combination therapy strategies. For example, we propose a phase I clinical trial to assess the safety and pharmacokinetics of HD in humans, followed by a phase II trial to evaluate its efficacy in hyperuricemia patients. We also suggest exploring the use of HD in combination with existing uric acid-lowering drugs to enhance therapeutic outcomes. These future directions have been added to the Discussion section under the heading "Future Research Directions."

(Please refer to Section 4.6 Limitations and Future Directions/3rd paragraph)

---

## [Decision Letter · Decision Letter 1]

Hederagenin's Uric Acid-Lowering Effects in Hyperuricemic Mice: Mechanistic Insights from Molecular Docking and In Vivo Analysis

PONE-D-24-51322R1

Dear Dr. Bai,

We’re pleased to inform you that your manuscript has been judged scientifically suitable for publication and will be formally accepted for publication once it meets all outstanding technical requirements.

Kind regards,

Sepiso K. Masenga, PhD

Academic Editor

PLOS ONE

Additional Editor Comments (optional):

Reviewers' comments:

Reviewer's Responses to Questions

**Comments to the Author**

Reviewer #1: All comments have been addressed

Reviewer #2: All comments have been addressed

2. Is the manuscript technically sound, and do the data support the conclusions?

Reviewer #1: (No Response)

Reviewer #2: Yes

3. Has the statistical analysis been performed appropriately and rigorously?

Reviewer #1: (No Response)

Reviewer #2: Yes

4. Have the authors made all data underlying the findings in their manuscript fully available?

Reviewer #1: (No Response)

Reviewer #2: Yes

5. Is the manuscript presented in an intelligible fashion and written in standard English?

Reviewer #1: (No Response)

Reviewer #2: Yes

Reviewer #1: (No Response)

Reviewer #2: All reviewer comments have been adequately addressed; as such, I am satisfied with recommending the acceptance of the manuscript.

**Do you want your identity to be public for this peer review?** For information about this choice, including consent withdrawal, please see our Privacy Policy

Reviewer #1: No

Reviewer #2: **Yes: ** Situmbeko Liweleya

---

## [Editor Report · Acceptance letter]

PONE-D-24-51322R1

PLOS ONE

Dear Dr. Bai,

I'm pleased to inform you that your manuscript has been deemed suitable for publication in PLOS ONE. Congratulations! Your manuscript is now being handed over to our production team.

Kind regards,

on behalf of

Prof. Sepiso K. Masenga

Academic Editor

PLOS ONE